# VISION: PROMPTING OCEAN VERTICAL VELOCITY RECONSTRUCTION FROM INCOMPLETE OBSERVATIONS

## ABSTRACT

Reconstructing subsurface ocean dynamics, such as vertical velocity fields, from incomplete surface observations poses a critical challenge in Earth science, a field long hampered by the lack of standardized, analysis-ready benchmarks. To systematically address this issue and catalyze research, we first build and release $KD48$, a high-resolution ocean dynamics benchmark derived from petascale simulations and curated with expert-driven denoising. Building on this benchmark, we introduce VISION, a novel reconstruction paradigm based on ***Dynamic Prompting*** designed to tackle the core problem of missing data in real-world observations. The essence of VISION lies in its ability to generate a *visual prompt* on-the-fly from *any available subset* of observations, which encodes both data availability and the ocean's physical state. More importantly, we design a State-conditioned Prompting module that efficiently injects this prompt into a universal backbone, endowed with geometry- and scale-aware operators, to guide its adaptive adjustment of computational strategies. This mechanism enables VISION to precisely handle the challenges posed by varying input combinations. Extensive experiments on the $KD48$ benchmark demonstrate that VISION not only substantially outperforms state-of-the-art models but also exhibits strong generalization under extreme data missing scenarios. By providing a high-quality benchmark and a robust model, our work establishes a solid infrastructure for ocean science research under data uncertainty. Our codes are available at: https://anonymous.4open.science/r/Anonymous_ICLR-8270.

## 1 INTRODUCTION

Ocean vertical velocity ($w$), a core driver of vertical mass and energy transport, plays a pivotal role in the global climate system, marine biogeochemical cycles, and ecosystem productivity (Burd, 2024; Liang et al., 2017; Denman & Gargett, 1995). Despite its fundamental importance, the direct observation of $w$ remains a long-standing bottleneck in oceanography. Its magnitude is typically several orders smaller than that of horizontal velocities, and it exhibits strong spatiotemporal variability, rendering large-scale, continuous, and reliable measurements technologically infeasible with current observational technologies (Muste et al., 2008). This fundamental paradox motivates the reconstruction of $w$ from more accessible sea surface observations, such as sea surface height (SSH) and sea surface temperature (SST) (Martin et al., 2023; Archambault, 2024). This direction is not only important for understanding internal ocean dynamics but also opens broad prospects for data-driven research in Earth science (Uchida et al., 2019; Martin et al., 2023).

To address the observational challenge, researchers have developed various methods for $w$ reconstruction. Traditional physics-based approaches, such as diagnostics based on quasi-geostrophic (QG) theory (Calkins, 2018; Held et al., 1995; Bishop & Thorpe, 1994), provide a theoretical foundation for understanding large-scale ocean circulation but rely on strong simplifying assumptions. These assumptions often fail in dynamically complex oceanic regions characterized by strong unbalanced flows and high Rossby numbers, limiting their applicability and accuracy (Warn et al., 1995; Fox-Kemper et al., 2019). Recently, deep learning (DL) methods have shown immense potential for the $w$ reconstruction task, leveraging their powerful ability to learn complex nonlinear mappings from large-scale numerical simulations (Zhu et al., 2023; He & Mahadevan, 2024). However, this emerging field faces **Dual Challenges**. ❶ *First, the rigidity in model design*. Existing DL models universally depend on a fixed and complete set of input variables. This requirement is at odds with the reality of

real-world observations, which are often incomplete due to various factors, thereby severely limiting the models' robustness and operational utility. ❷ *Second, the scarcity of high-quality benchmark datasets.* Current research often relies on disparate, small-scale datasets processed in-house, which not only imposes a significant data engineering burden on researchers but also impedes fair comparisons between different methods and hinders reproducible research within the community. These intertwined challenges have become a critical bottleneck constraining the advancement of the field.

To systematically address these dual challenges, we propose this work. ***First, to tackle the data scarcity problem, we build and publicly release the Kuroshio-Dynamics-48 ($KD48$) dataset***. Derived from petascale, high-resolution simulations and curated with expert-driven dynamical signal filtering, $KD48$ is the first large-scale, analysis-ready benchmark specifically designed for ocean dynamics reconstruction under data uncertainty. ***Second, to address the model rigidity problem, we propose*** VISION, ***a novel reconstruction paradigm based on Dynamic Prompting (Wu et al., 2024a), built upon the*** $KD48$ ***benchmark***. The core idea of this paradigm is to train a universal, promptable backbone network. We design a state-conditioned prompting module that generates a dynamic prompt on-the-fly from any available subset of observations, encoding both data availability and the physical state of the ocean. This prompt is then efficiently injected into a custom-designed backbone, featuring geometry- and scale-aware operators, to guide the adaptive adjustment of its computational strategies, thereby precisely handling the challenges posed by varying input combinations.

The main contributions of this paper are summarized as follows:

❶ We construct and release the Kuroshio-Dynamics-48 (*KD48*), the first high-resolution, analysis-ready benchmark dataset specifically focused on $w$ reconstruction under data uncertainty. It fills a critical gap in the field and provides a standardized platform for fair model evaluation.

❷ We propose VISION, a novel prompt-driven framework that systematically addresses the performance degradation caused by dynamic input unavailability in ocean reconstruction for the first time, significantly enhancing model robustness and practical utility.

❸ Extensive experiments on the $KD48$ benchmark demonstrate that VISION substantially outperforms various state-of-the-art baselines under diverse data missing scenarios, showcasing its excellent performance and generalization capabilities.

Above of all, by providing a high-quality benchmark and a robust model, our work establishes a solid infrastructure for real-world ocean science applications and offers a new paradigm for other scientific computing domains facing similar data uncertainty challenges.

## 2 RELATED WORK

**Ocean Vertical Velocity Reconstruction.**    Reconstructing ocean vertical velocity ($w$) is a long-standing challenge in physical oceanography (Mahadevan et al., 2020; Röhrs et al., 2023). Classical methods, predominantly based on quasi-geostrophic (QG) theory, diagnose $w$ by solving the Omega equation under assumptions of dynamical balance (Isern-Fontanet et al., 2006; Lapeyre & Klein, 2006). While foundational, these physics-based approaches have inherent limitations in regions dominated by strong, ageostrophic submesoscale dynamics. Recently, deep learning (DL) has emerged as a powerful data-driven alternative, using neural networks to learn the complex, nonlinear relationships between sea surface observables and subsurface $w$ from high-resolution numerical simulations (Zhu et al., 2023; He & Mahadevan, 2024). These models demonstrate significant improvements in accuracy over traditional methods. *However, a common thread among existing DL approaches is their reliance on a fixed, complete set of input variables.* This design assumes that all prescribed inputs are consistently available, a condition rarely met in real-world observational scenarios (Glenn et al., 2000; Zeng et al., 2020). In contrast, our work focuses on developing a model that is robust to the dynamic availability of input variables.

**Scientific Machine Learning with Incomplete Data.**    The challenge of incomplete or missing data is pervasive across scientific machine learning domains, from weather forecasting (Bi et al., 2023; Zhang et al., 2023; Wu et al., 2024a;b; Gao et al., 2025b;a; Wu et al., 2025b), spatiotemporal data mining (Raonic et al., 2023; Wu et al., 2023b; 2024c; Wang et al.; Wu et al.; Li et al., 2025; Wu et al., 2025c;a), to biomedical imaging (Webb, 2022; Tempany & McNeil, 2001; Acharya et al., 1995). Traditional approaches often involve a pre-processing step of data imputation, using

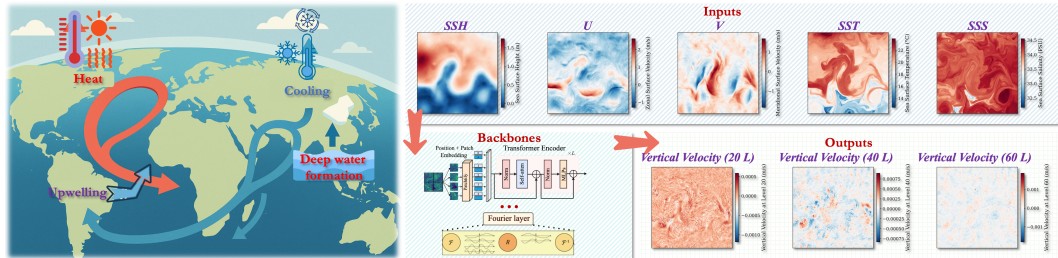

Figure 1: **Overview of the** $KD48$ **benchmark.** (***Left***) The scientific motivation: reconstructing vertical velocity ($w$), a key driver of the global ocean's thermohaline circulation. (***Right***) The corresponding supervised learning task, which involves mapping five observable sea surface variables to the subsurface vertical velocity at three different depths using deep learning models.

methods ranging from simple interpolation to more sophisticated generative models like GANs (Pan et al., 2020; Hussein et al., 2020). While effective for certain tasks, these methods treat imputation and the downstream scientific task as two separate problems, which can introduce artifacts and propagate errors (Adhikari et al., 2022; Luo et al., 2018; Cao et al., 2018). More recent works, particularly in the graph neural network (GNN) domain, are inherently more flexible to missing nodes or features (Guskov et al., 2002; Huang & Yang, 2021; Fan et al., 2019). Our approach differs fundamentally from these paradigms. Instead of explicitly *filling in* missing data, VISION adopts an end-to-end strategy that learns to perform optimally with *whatever data is present*. It achieves this by conditioning its computations directly on data availability, a more direct and potentially more robust strategy than multi-stage imputation-then-prediction pipelines.

**Prompt Learning in Deep Learning.** Prompt learning has recently revolutionized the field of artificial intelligence, emerging as a powerful paradigm for adapting large pre-trained models to a wide array of downstream tasks (Zamfirescu-Pereira et al., 2023; Guo et al., 2024; Mizrahi et al., 2024; Khattak et al., 2025). Initially popularized by Large Language Models (LLMs) (Zhao et al., 2023; Kirchenbauer et al., 2023; Minaee et al., 2024), the core idea is to guide a model's behavior using task-specific instructions, or *prompts* (Wu et al., 2024a; Khattak et al., 2025; Pan et al., 2024; Ma et al., 2025), rather than updating its weights. This concept has been successfully extended to vision-language models (VLMs), where textual or visual prompts are used to steer tasks like image segmentation and object detection (Zang et al., 2025; Du et al., 2022). While transformative, the application of prompt-based learning to complex physical systems and scientific computing remains a nascent area of research. ***To our knowledge, our work is the first to systematically apply the prompting paradigm to the challenge of ocean dynamics reconstruction.*** We introduce a novel form of conditioning: a state-and-availability prompt that encodes both the physical context and the meta-information of the data, thereby opening a new avenue for applying prompt-based learning to scientific problems characterized by data uncertainty.

## 3 THE $KD48$ BENCHMARK

Reconstructing ocean vertical velocity ($w$) is a critical challenge in Earth science, where progress has long been hampered by the lack of standardized, analysis-ready benchmarks. As shown in the physical schematic in Figure 1 (left), vertical velocity is the core engine driving the global climate system's "conveyor belt"—the thermohaline circulation. It governs the global vertical transport of heat and mass through processes like deep water formation and upwelling. However, due to its faint signal and the difficulty of direct observation, reconstructing $w$ from more accessible surface data (e.g., SSH, SST) has become a vital scientific task. To systematically address this issue and catalyze research, we construct and release the **Kuroshio-Dynamics-48** ($KD48$) benchmark.

The data engineering pipeline for $KD48$, shown in Figure 2(a), is designed to transform a complex scientific problem into a well-posed machine learning task. We source our data from the petascale LLC4320 ocean simulation, selecting the Kuroshio Extension region, an area known for its highly complex dynamics. The benchmark explicitly frames the reconstruction task as a mapping from a multi-channel 2D surface observation field to a target 3D subsurface physical field. The ***Inputs*** consist of five sea surface fields observations: Sea Surface Height (SSH), Sea Surface Temperature

(SST), Sea Surface Salinity (SSS), and zonal (U) and meridional (V) surface velocities. The **Outputs** are the $w$ at three distinct subsurface depths (Level 20, 40, and 60).

A core contribution of this benchmark lies in the meticulous curation of the ground truth. Using the raw vertical velocity ($w_{\text{raw}}$) from the simulation directly poses an ill-posed learning problem, as it includes high-frequency noise (e.g., internal tides) that is only weakly coupled, physically, to the surface inputs. To address this, we design and apply a dynamical signal filter. This filter isolates the signal component ($w^*$) that is dynamically consistent with the evolution of surface eddies and fronts from the raw signal. This refined signal, $w^*$, serves as the final learning target. This critical step ensures a well-defined physical mapping between the inputs and outputs, thereby guiding models to learn genuine physical dynamics rather than fitting spurious noise.

Ultimately, $KD48$ provides a full year of hourly, high-resolution (1/48°) data, constituting a large-scale, physically consistent, and challenging platform.

## 4 METHOD

### 4.1 PROBLEM FORMULATION AND DESIGN PRINCIPLES

Let $\mathcal{V} = \{v_i\}_{i=1}^N$ denote a universe of $N$ potentially available sea surface variables. An observation at any given time is defined by a subset of variables $\mathcal{S} \subseteq \mathcal{V}$, corresponding to a multi-channel input tensor $\mathbf{X}_\mathcal{S} \in \mathbb{R}^{|\mathcal{S}| \times H \times W}$, where $H$ and $W$ are the spatial dimensions (height and width). Our core task is to learn a single, parameterized mapping $f_\theta : \mathcal{X} \to \mathcal{W}$ that projects any tensor $\mathbf{X}_\mathcal{S}$ from the input space $\mathcal{X} = \bigcup_{\mathcal{S} \subseteq \mathcal{V}} \mathbb{R}^{|\mathcal{S}| \times H \times W}$ to a vertical velocity field $\mathbf{w}$ in the target space $\mathcal{W} = \mathbb{R}^{C \times H \times W}$, where C denotes the number of channels for multi-layer $w$. This mapping is governed by a *universal* set of parameters $\theta$ that must remain effective across possible non-empty subsets $\mathcal{S}$ without retraining. Formally, our objective is to find the optimal parameters $\theta^*$ that minimize the expected loss over all possible input subsets and data samples:

$$\theta^* = \arg\min_\theta \; \mathbb{E}_{(\mathcal{S}, \mathbf{X}_\mathcal{S}, \mathbf{w}) \sim \mathcal{D}} \left[ \mathcal{L} \left( f_\theta(\mathbf{X}_\mathcal{S}), \mathbf{w} \right) \right], \tag{1}$$

where $\mathcal{D}$ is the true data-generating distribution and $\mathcal{L}$ is a suitable loss function. The central challenge lies in designing a function $f_\theta$ that can handle a variable-dimensional, combinatorially large input space while extracting consistent predictive features for the target $\mathbf{w}$.

➡ **Design Principles.** To address the challenge defined in Eq. equation 1, our model design is shaped by two fundamental principles: *universality* and *state-conditioned adaptivity*. Universality mandates that the model must function for *any* input subset $\mathcal{S}$ without requiring architectural surgery or retraining, which implies the need for a front-end that can canonicalize arbitrary variable combinations into a fixed-dimensional internal representation. However, universality alone is insufficient. An ideal model must also adapt its computational strategy based on the current context. This adaptivity should be two-fold: it must be *availability-aware*, dynamically altering its computational paths based on which variables are present or absent, and it must be *state-aware*, conditioning its behavior on the macroscopic dynamical state of the ocean reflected in the observations. To realize these principles, we propose `VISION`, an end-to-end framework composed of an *Adaptive Observation Embedder* and a *Geometry-Scale Aware Operator*, as shown in Figure 2.

### 4.2 ADAPTIVE EMBEDDING AND PROMPT-GUIDED ADAPTATION

➡ **Universal Observation Adapter.** The adaptive capability of `VISION` begins at its front-end, the Adaptive Observation Embedder (AOE), which first canonicalizes any observation subset $\mathbf{X}_\mathcal{S} \in \mathbb{R}^{|\mathcal{S}| \times H \times W}$ via a Universal Observation Adapter (UOA). The UOA employs a shared, availability-aware linear operator $\phi_{\text{UOA}}$ to map the variable-dimensional input to a base feature tensor $\mathbf{Z}_0 \in \mathbb{R}^{C_b \times H \times W}$ with a fixed channel dimension:

$$\mathbf{Z}_0 = \phi_{\text{UOA}}(\mathbf{X}_\mathcal{S}) = \left( W^{(S)} \otimes \mathbf{I}_{1 \times 1} \right) \mathbf{X}_\mathcal{S}, \quad W^{(S)} \in \mathbb{R}^{S \times C_b}, \tag{2}$$

where the operator $W^{(S)}$ corresponds to a learnable projection matrix dynamically chosen according to the input channel $S$, while $\otimes \mathbf{I}_{1 \times 1}$ constrains the mapping to a $1 \times 1$ convolutional kernel. This design harmonizes heterogeneous input modalities into a consistent $C_b$-dimensional feature space, thereby enabling subsequent modules to operate on aligned representations.

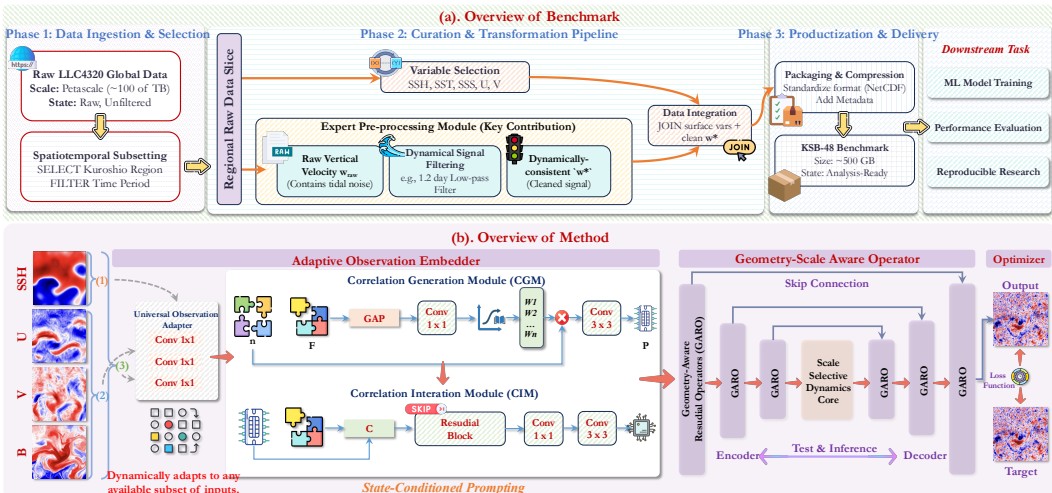

Figure 2: **Overview of the $KD48$ benchmark construction pipeline and the `VISION` model framework. (a)** The $KD48$ benchmark is constructed by first subsetting the Kuroshio region from petascale LLC4320 data, followed by an expert pre-processing module (a key contribution) that filters raw vertical velocity ($w_{\mathrm{raw}}$) into a dynamically consistent target ($w^*$), and finally integrating it with surface variables into an analysis-ready format. **(b)** The `VISION` model consists of two main components: an Adaptive Observation Embedder that generates a dynamic prompt $P$ from any available input subset via a *SCP* mechanism, and a prompt-conditioned ***Geometry-Scale Aware Operator*** that reconstructs the final vertical velocity field.

⇛ **Adaptation via State-Conditioned Prompting.** After obtaining the canonicalized features $\mathbf{Z}_0$, the model's core adaptive capability is realized through the *State-Conditioned Prompting (SCP)* module. This module generates a dynamic spatial prompt $\mathbf{P}$, which serves as an **adaptation signal** providing precise information about the current observational context to the downstream reconstruction operator. The process begins by compressing $\mathbf{Z}_0$ into a state vector $\mathbf{e}$, which, concatenated with the availability mask $\mathbf{m}$, is mapped by an MLP $\phi_{\mathrm{mixer}}$ to a set of mixing weights $\boldsymbol{\alpha} \in \Delta^{K-1}$:

$$\boldsymbol{\alpha} = \mathrm{Softmax}(\phi_{\mathrm{mixer}}(\mathbf{e})). \tag{3}$$

These weights are used to form a linear combination of a learnable codebook of prompt templates, $\mathcal{C}_{\mathbf{P}} = \{\mathbf{P}_k\}_{k=1}^K$, yielding the final dynamic prompt $\mathbf{P}$:

$$\mathbf{P}(\mathbf{e}) = \mathrm{Conv}_{3\times3}\left(\mathcal{U}\left(\sum_{k=1}^K \alpha_k \mathbf{P}_k\right)\right), \tag{4}$$

where $\mathcal{U}$ is an upsampling operator to align the prompt with the latent feature. The resulting prompt $\mathbf{P}$ is then deeply fused with the base features $\mathbf{Z}_0$ via a residual **Prompt Interaction Module**, $\Gamma_{\mathrm{int}}$, to produce the conditioned feature tensor $\mathbf{Z}_1$:

$$\mathbf{Z}_1 = \Gamma_{\mathrm{int}}(\mathbf{Z}_0, \mathbf{P}) = \mathrm{Conv}_{3\times3}(\mathcal{Q}([\mathbf{Z}_0; \mathbf{P}])), \tag{5}$$

where $\mathcal{Q}$ denotes the resudial block. In this manner, the prompt $\mathbf{P}$ guides the GSAO operator to adapt its behavior, enabling it to select optimal computational paths based on the input composition and the ocean state.

**Theoretical Justification.** The effectiveness of this prompt-guided adaptation, particularly its ability to leverage multi-variable inputs, is supported by information theory. The following lemma formalizes why integrating a richer set of observational variables fundamentally improves the reconstruction task. A detailed proof is provided in Appendix B.

**Lemma 1 (Monotonicity of Observational Information)** *(Shannon, 1948; MacKay, 2003) Let $\mathbf{w}$ be the target field to be reconstructed. Let $\mathcal{S}_1$ and $\mathcal{S}_2$ be two sets of observable variables such that $\mathcal{S}_1 \subset \mathcal{S}_2$. Let $\mathbf{X}_{\mathcal{S}_1}$ and $\mathbf{X}_{\mathcal{S}_2}$ denote the corresponding noise-free observations. The conditional entropy (i.e., the remaining uncertainty) of $\mathbf{w}$ given these observations satisfies:*

$$H(\mathbf{w}|\mathbf{X}_{\mathcal{S}_2}) \leq H(\mathbf{w}|\mathbf{X}_{\mathcal{S}_1}) \tag{6}$$

*In a coupled physical system like the ocean, where different variables provide unique, complementary constraints, this inequality is strict. This implies that incorporating more observational variables strictly reduces the intrinsic uncertainty of the reconstruction task.*

This lemma provides a theoretical guarantee that the solution space becomes more constrained as more variables are observed. Our state-conditioned prompt $\mathbf{P}$ serves as the mechanism to effectively communicate these tighter constraints to the model backbone, thereby steering the reconstruction towards a more accurate and physically consistent solution.

### 4.3 GEOMETRY-SCALE AWARE OPERATOR

*The Geometry-Scale Aware Operator* (GSAO) serves as the backbone of `VISION`, functioning as an Encoder-Decoder architecture that takes the conditioned features $\mathbf{Z}_1$ as input. The operator is specifically designed to efficiently capture the multi-scale and geometric features inherent in ocean dynamics, with its core composed of two specialized residual modules: the *Geometry-Aware Residual Operators (GARO)* and the *Scale-Selective Dynamics Core (SSDC)*.

⇛ **Geometry-Aware Residual Operators.** To align the model's computation with the geometry of flow fields (e.g., eddies and fronts), we employ a residual operator, GARO, based on deformable convolutions. Unlike standard convolutions, GARO learns an additional 2D spatial offset $\Delta \mathbf{p} = \mathcal{O}(\mathbf{F_g})$ for each sampling point of the kernel, determined by the input features $\mathbf{F_g}$. This allows the sampling locations to dynamically focus on the most informative regions of the feature, such as areas with high gradients along fronts. The process is formalized as:

$$\mathrm{GARO}(\mathbf{F_g}) = \mathrm{Conv}\big(\mathrm{Sample}(\mathbf{F_g}, \mathbf{p}_0 + \Delta \mathbf{p})\big) + \mathbf{F_g}, \tag{7}$$

where $\mathbf{p}_0$ denotes the original regular grid. This mechanism makes the model's receptive fields with geometry-awareness, significantly enhancing its ability to capture fine-grained physical structures.

⇛ **Scale-Selective Dynamics Core.** To adaptively handle the multi-scale nature of ocean dynamics, we design the SSDC module, located at the bottleneck of the encoder. This module employs a selective kernel operator that dynamically fuses the outputs of a set of convolutional kernels with varying sizes ($k_r \in \mathcal{R}$). A squeeze-and-excitation module, $\mathcal{A}$, generates a set of mixing weights $\boldsymbol{\omega}^\ell$ based on the input features $\mathbf{F}^\ell$:

$$\mathbf{F_s}^{\ell+1} = \Psi^\ell \left( \sum_{r \in \mathcal{R}} \omega_r^\ell \cdot \mathrm{Conv}_{k_r}(\mathbf{F_s}^\ell) \right), \quad \boldsymbol{\omega}^\ell = \mathrm{softmax}\big(\mathcal{A}(\mathbf{F_s}^\ell)\big), \tag{8}$$

where $\Psi^\ell$ is a residual block. This mechanism realizes dynamic scale-selection, enabling the model to better capture complex dynamics ranging from large-scale eddies to small-scale filaments.

⇛ **Decoding, Prediction, and Optimization.** The decoder of the GSAO fuses features from different encoder levels via skip connections to generate a high-resolution feature map $\mathbf{F}^{\mathrm{HR}}$. Subsequently, a $1 \times 1$ convolutional layer projects this feature map to a single channel, yielding the final reconstructed vertical velocity field $\hat{\mathbf{w}} \in \mathbb{R}^{1 \times H \times W}$. The entire `VISION` model is trainable end-to-end. Given a training sample $(\mathbf{X}_\mathcal{S}, \mathbf{w})$, we optimize the model parameters $\theta$ by minimizing the Smooth $L_1$ Loss between the prediction $\hat{\mathbf{w}}$ and the ground-truth $\mathbf{w}$. This loss function combines the robustness of L1 loss to outliers with the stability of $L_2$ loss near zero. The optimization objective is defined as:

$$\mathcal{L}(\theta) = \mathcal{L}_{\mathrm{smooth\ L1}}(\hat{\mathbf{w}}, \mathbf{w}). \tag{9}$$

By minimizing this objective over a large-scale dataset (such as $KD48$) that encompasses a diverse range of available variable subsets $\mathcal{S}$, `VISION` learns to robustly perform reconstruction via dynamic prompting, rather than merely memorizing specific input-output patterns.

## 5 EXPERIMENT

To comprehensively evaluate the performance of `VISION` in ocean vertical velocity reconstruction task and validate its effectiveness in real-world applications, where some surface variables may be missing, we design a series of rigorous experiments. All experiments are conducted on 8 NVIDIA 40GB-A100 GPUs.

Table 1: Reconstruction performance comparison on various subterranean layers. We benchmark our method, `VISION`, against four categories of baselines: Operator Learning Models (OLM), Computer `VISION` Backbones (CVB), Spatiotemporal Models (STM), and Domain-specific Models (DSM). All models are evaluated on RMSE (↓), MAE (↓), and PCC (↑); lower RMSE/MAE and higher PCC indicate better performance. **Bold** denotes the best result, and a single underline indicates the second-best. Our model, `VISION` (IO), is trained with incomplete observations, while all baselines use complete observations (CO).

| MODEL | SUBTERRANEAN LAYERS | | | | | | | | |
| --- | --- | --- | --- | --- | --- | --- | --- | --- | --- |
| | 20 LAYERS | | | 40 LAYERS | | | 60 LAYERS | | |
| | RMSE (↓) | MAE (↓) | PCC (↑) | RMSE (↓) | MAE (↓) | PCC (↑) | RMSE (↓) | MAE (↓) | PCC (↑) |
| **OPERATOR LEARNING MODELS (OLM)** | | | | | | | | | |
| 🪅 FNO | $7.216\times10^{-5}$ | $5.265\times10^{-5}$ | 0.240 | $1.269\times10^{-4}$ | $9.624\times10^{-5}$ | 0.344 | $1.561\times10^{-4}$ | $1.219\times10^{-4}$ | 0.234 |
| 🪅 CNO | $5.668\times10^{-5}$ | $\underline{4.021\times10^{-5}}$ | 0.643 | $1.109\times10^{-4}$ | $8.391\times10^{-5}$ | 0.559 | $1.497\times10^{-4}$ | $1.171\times10^{-4}$ | 0.352 |
| 🪅 LSM | $\underline{5.651\times10^{-5}}$ | $4.034\times10^{-5}$ | $\underline{0.645}$ | $1.049\times10^{-4}$ | $7.938\times10^{-5}$ | 0.620 | $1.355\times10^{-4}$ | $1.055\times10^{-4}$ | 0.529 |
| **COMPUTER VISION BACKBONES (CVB)** | | | | | | | | | |
| 📷 U-NET | $6.426\times10^{-5}$ | $4.568\times10^{-5}$ | 0.500 | $1.062\times10^{-4}$ | $8.029\times10^{-5}$ | 0.609 | $1.360\times10^{-4}$ | $1.058\times10^{-4}$ | 0.526 |
| 📷 RESNET | $5.759\times10^{-5}$ | $4.086\times10^{-5}$ | 0.630 | $1.123\times10^{-4}$ | $8.455\times10^{-5}$ | 0.554 | $1.464\times10^{-4}$ | $1.142\times10^{-4}$ | 0.413 |
| **SPATIOTEMPORAL MODELS (STM)** | | | | | | | | | |
| ⊞ SIMVP | $5.765\times10^{-5}$ | $4.107\times10^{-5}$ | 0.627 | $\underline{1.044\times10^{-4}}$ | $\underline{7.903\times10^{-5}}$ | $\underline{0.625}$ | $\underline{1.346\times10^{-4}}$ | $\underline{1.048\times10^{-4}}$ | $\underline{0.539}$ |
| **DOMAIN-SPECIFIC MODELS (DSM)** | | | | | | | | | |
| ⚙ DNN | $6.902\times10^{-5}$ | $4.929\times10^{-5}$ | 0.413 | $1.273\times10^{-4}$ | $9.503\times10^{-5}$ | 0.358 | $1.572\times10^{-4}$ | $1.224\times10^{-4}$ | 0.200 |
| 🏆 `VISION` (IO) | $\mathbf{5.519\times10^{-5}}$ | $\mathbf{3.935\times10^{-5}}$ | $\mathbf{0.667}$ | $\mathbf{1.034\times10^{-4}}$ | $\mathbf{7.833\times10^{-5}}$ | $\mathbf{0.634}$ | $\mathbf{1.335\times10^{-4}}$ | $\mathbf{1.039\times10^{-4}}$ | $\mathbf{0.549}$ |

Figure 3: Qualitative comparison of vertical velocity ($w$) reconstruction at three subterranean depths (W20, W40, W60). The figure compares the outputs of our proposed model `VISION` (Ours) against several state-of-the-art baselines, including CNO, FNO, UNet, ResNet, SimVP, and the domain-specific DDN, with the Ground Truth shown on the far right. While most baselines either produce overly smoothed results (e.g., FNO) or fail to capture coherent structures (e.g., DDN), our model successfully reconstructs the complex, multi-scale turbulent features, demonstrating superior performance in capturing the physical dynamics.

## 5.1 DATA AND BASELINES

We use the proposed $KD48$ benchmark to conduct analysis of ocean vertical velocity reconstruction. Specifically, we use the hourly snapshots of Sea Surface Height (SSH), Buoyancy (B), which is calculated by Sea Surface Temperature (SST) and Sea Surface Salinity (SSS), zonal (U) and meridional (V) surface velocities, and depth level vertical velocity w at 20, 40, and 60. In summary, we use 8000 samples for training, 500 samples for validating, and 1500 samples for testing. We compare our proposed `VISION` with 4 types of baselines, which includes operator learning models (FNO (Li et al., 2021), CNO (Raonic et al., 2023), and LSM (Wu et al., 2023a)), computer vision backbones (UNet (Ronneberger et al., 2015) and ResNet (He et al., 2016)), spatiotemporal model (SimVP (Tan et al., 2025)), and domain-specific model for w reconstruction (DDN (Zhu et al., 2023)). The baseline models are trained using the complete observation, and our `VISION` is trained using random observation, which randomly selects input from incomplete observation or complete observation.

## 5.2 COMPARISON WITH STATE-OF-THE-ART METHODS

We use three metrics, Root Mean Square Error (RMSE), Mean Absolute Error (MAE), and Pearson Correlation Coefficient (PCC) to evaluate the reconstruction performance of different methods. More details can be found in E.2. As shown in Table 1, we report the average results for 1500 samples. Although VISION is trained in incomplete observation (IO) settings, it still achieves competitive performance compared to state-of-the-art baselines. In contrast, baseline models rely on complete observation (CO), which restricts their applicability in real-world scenarios where certain variables are inevitably missing. Furthermore, as illustrated in Figure 3, the reconstruction performance of VISION are in closer agreement with the ground truth.

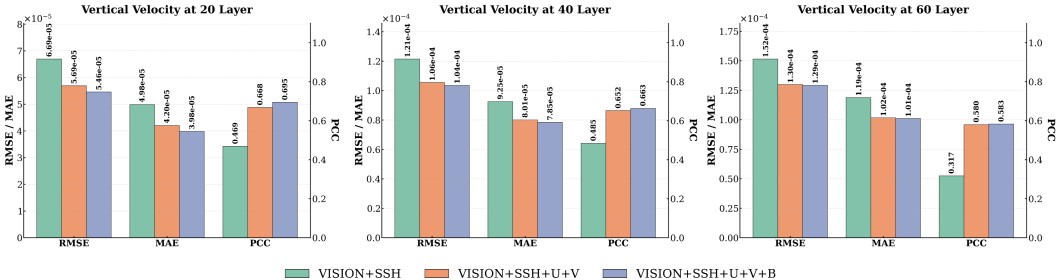

Figure 4: **Quantitative Validation of Dynamic Prompting.** Reconstruction performance of VISION at three depths (20, 40, 60) improves as more input variables (SSH, U, V, B) are provided. The consistent reduction in RMSE/MAE and increase in PCC validate VISION's ability to adaptively leverage available observational data.

**From a quantitative perspective.** Table 1 demonstrates the consistent and superior performance of our proposed VISION model. Across all three subterranean depths (20, 40, and 60 layers), VISION achieves the best results on all evaluation metrics: the lowest RMSE and MAE, and the highest PCC. This achievement is particularly noteworthy because VISION is trained under the more challenging and realistic scenario of IO, whereas all baseline models are trained with the advantage of CO. This indicates that our method not only reaches a higher level of accuracy but also possesses superior robustness and practical value for real-world applications.

**From a qualitative standpoint.** The visual results in Figure 3 provide a compelling confirmation of VISION's capabilities. The Ground Truth images are characterized by complex, multi-scale turbulent structures, including sharp fronts and fine filaments. In comparison, baseline models show significant deficiencies. For instance, FNO produces overly smoothed fields that lose almost all fine-scale details, while DDN fails almost completely, capturing only a few sparse, high-intensity spots. Although

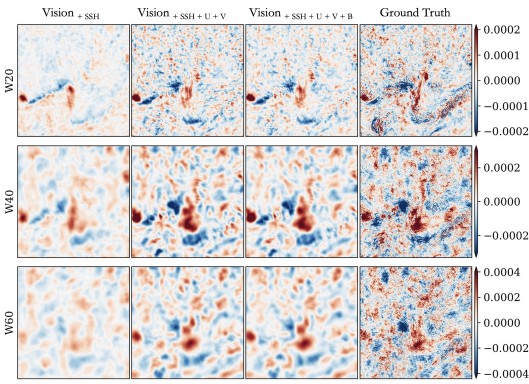

Figure 5: **Qualitative Comparison of Dynamic Prompting Reconstruction**. This figure demonstrates the progressive improvement of VISION's reconstruction of vertical velocity at three depths (W20, W40, W60). As more input variables are provided from only SSH, to including surface velocities (U+V), and finally B the reconstructed fields become increasingly detailed and more closely resemble the Ground Truth.

DDN is a domain-specific method for $w$ reconstruction, it will produce unsatisfactory performance on more challenging $KD48$ benchmark when it deviates from the relatively simple ideal data used in their paper. Other models like CNO, UNet, and ResNet capture large-scale patterns but still appear blurry and fail to resolve the smaller intricate structures. In stark contrast, the output from VISION (Ours) shows a remarkable visual fidelity to the ground truth. It accurately reconstructs not only the large-scale upwelling (red) and downwelling (blue) zones but also successfully resolves many of the fine, filamentary details, presenting a physically coherent and detailed velocity field that is far superior to all baselines.

Table 2: **Ablation study of** `VISION`**'s key components on the** $KD48$ **benchmark.** We evaluate the impact of each component under different observation settings. Performance degradation in ablated models highlights their necessity. Best results are in **bold**.

| MODEL VARIANT | SUBTERRANEAN LAYERS | | | | | | | | |
| --- | --- | --- | --- | --- | --- | --- | --- | --- | --- |
| | 20 LAYER | | | 40 LAYER | | | 60 LAYER | | |
| | RMSE↓ | MAE↓ | PCC↑ | RMSE↓ | MAE↓ | PCC↑ | RMSE↓ | MAE↓ | PCC↑ |
| *Incomplete Observation (SSH)* | | | | | | | | | |
| ⊖ *w/o SCP* | $7.359\times10^{-5}$ | $5.356\times10^{-5}$ | 0.280 | $1.335\times10^{-4}$ | $1.018\times10^{-4}$ | 0.241 | $1.604\times10^{-4}$ | $1.252\times10^{-4}$ | 0.132 |
| ★ `VISION` | $\mathbf{6.783\times10^{-5}}$ | $\mathbf{4.960\times10^{-5}}$ | **0.433** | $\mathbf{1.195\times10^{-4}}$ | $\mathbf{9.059\times10^{-5}}$ | **0.463** | $\mathbf{1.507\times10^{-4}}$ | $\mathbf{1.178\times10^{-4}}$ | **0.306** |
| *Incomplete Observation (SSH U V)* | | | | | | | | | |
| ⊖ *w/o SCP* | $6.315\times10^{-5}$ | $4.528\times10^{-5}$ | 0.557 | $1.160\times10^{-4}$ | $8.762\times10^{-5}$ | 0.510 | $1.493\times10^{-4}$ | $1.166\times10^{-4}$ | 0.345 |
| ★ `VISION` | $\mathbf{5.849\times10^{-5}}$ | $\mathbf{4.205\times10^{-5}}$ | **0.640** | $\mathbf{1.065\times10^{-4}}$ | $\mathbf{8.036\times10^{-5}}$ | **0.618** | $\mathbf{1.326\times10^{-4}}$ | $\mathbf{1.032\times10^{-4}}$ | **0.545** |
| *Complete Observation (SSH U V B)* | | | | | | | | | |
| ⊖ *w/o SCP* | $6.076\times10^{-5}$ | $4.330\times10^{-5}$ | 0.593 | $1.157\times10^{-4}$ | $8.739\times10^{-5}$ | 0.513 | $1.500\times10^{-4}$ | $1.171\times10^{-4}$ | 0.336 |
| ⊖ *w/o GSAO* | $7.359\times10^{-5}$ | $5.356\times10^{-5}$ | 0.280 | $1.335\times10^{-4}$ | $1.018\times10^{-4}$ | 0.241 | $1.604\times10^{-4}$ | $1.252\times10^{-4}$ | 0.132 |
| ★ `VISION` | $\mathbf{5.605\times10^{-5}}$ | $\mathbf{3.975\times10^{-5}}$ | **0.668** | $\mathbf{1.045\times10^{-4}}$ | $\mathbf{7.872\times10^{-5}}$ | **0.630** | $\mathbf{1.316\times10^{-4}}$ | $\mathbf{1.025\times10^{-4}}$ | **0.550** |

### 5.3 DYNAMIC PROMPTING RECONSTRUCTION EVALUATION

We evaluate `VISION`'s dynamic prompting mechanism through a series of experiments. The core advantage of this mechanism is its ability to adaptively tailor its reconstruction strategy to any available combination of inputs. This design philosophy aligns perfectly with our theoretical foundation, ***Lemma*** 1, which posits from an information-theoretic standpoint that more observational information effectively reduces the inherent uncertainty of the reconstruction task.

Our experimental results provide strong empirical support for this theory from both quantitative and qualitative perspectives. Quantitatively, as shown in Figure 4, the model's reconstruction error consistently decreases and its correlation with the ground truth steadily improves as input variables are augmented from only SSH to the full set including U, V and B. Qualitatively, this performance gain is mirrored by a remarkable enhancement in visual fidelity (Figure 5). The model's output evolves from an initially blurry, large-scale approximation to a highly detailed field that accurately resolves complex eddies and fronts, ultimately achieving a close match with the Ground Truth when all inputs are used. ***This progression from blurry to realistic vividly demonstrates how `VISION` translates theoretical information gain into more physically consistent and detailed reconstructions.***

### 5.4 ABLATION STUDY

To verify the effectiveness of the proposed method, as shown in Table 2, we conduct detailed ablation experiments. The model variants and `VISION` are trained using random observation, which randomly selects input from incomplete observation or complete observation. During inference, we report the average performance over 1,000 samples across three observation scenarios: **Incomplete Observation (SSH)**, **Incomplete Observation (SSH, U, V)**, and **Complete Observation (SSH, U, V, B)**. `VISION` W/O SCP represents that we remove State-Conditioned Prompting (SCP). `VISION` W/O GSAO means that we remove the Geometry-Scale Aware Operator (GSAO). Experimental results show that the lack of any component will degrade the performance of, which proves the effectiveness of the proposed method. More importantly, the introduced promoting strategy is vital to the real-world $w$ reconstruction, where the observations are often incomplete.

## 6 CONCLUSION

This work introduces `VISION`, a framework for reconstructing ocean vertical velocity from incomplete observations. Using a novel Dynamic Prompting mechanism, `VISION` adaptively processes any subset of available surface variables, overcoming the brittleness of traditional models to missing data. To facilitate research, we also construct and release the $KD48$ benchmark, a large-scale, high-quality dataset. Extensive experiments on $KD48$ demonstrate that `VISION` substantially outperforms state-of-the-art models while exhibiting exceptional robustness and generalization across diverse data-missing scenarios. Our work thus provides a robust adaptive model and a standardized benchmark, establishing a new paradigm for handling data uncertainty in scientific computing.

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

## A   THE USE OF LARGE LANGUAGE MODELS (LLMS)

LLMs were not involved in the research ideation or the writing of this paper.

## B   PROOF OF LEMMA 1

**Lemma 1** *Let $\mathbf{w}$ be the target random variable representing the field to be reconstructed. Let $S_1$ and $S_2$ be two sets of observable variables such that $S_1 \subset S_2$. Let $\mathbf{X}_{S_1}$ and $\mathbf{X}_{S_2}$ denote the random variables for the corresponding observations. The conditional entropy of $\mathbf{w}$ given these observations satisfies:*

$$H(\mathbf{w}|\mathbf{X}_{S_2}) \leq H(\mathbf{w}|\mathbf{X}_{S_1}) \tag{10}$$

**Proof B.1** *The proof is structured in five steps. We first define the necessary concepts, then derive the main result by leveraging the non-negativity of conditional mutual information, and finally discuss the condition for equality.*

**1.  Definitions and Setup**   *To establish the proof, we first define the key symbols and concepts from information theory. These are summarized in Table 3. The arguments extend from discrete to continuous variables by replacing summations with integrals.*

Table 3: Summary of key symbols and definitions used in the proof.

| Symbol | Description | Conceptual Definition |
|--------|-------------|----------------------|
| $\mathbf{w}$ | Target Field | The random variable for the field to be reconstructed. |
| $\mathbf{X}_{\mathcal{S}}$ | Observations | The random variable for observations from variable set $\mathcal{S}$. |
| $H(A)$ | Entropy | Measures the average uncertainty of a random variable $A$. |
| $H(A|B)$ | Conditional Entropy | Remaining uncertainty of $A$ given that $B$ is known. |
| $I(A;B|C)$ | Conditional Mutual Info. | Mutual information between $A$ and $B$ given $C$. |
| $D_{KL}(P \parallel Q)$ | KL Divergence | A measure of how one probability distribution $P$ diverges from $Q$. |

*Let $\mathcal{S}_{add} = \mathcal{S}_2 \backslash \mathcal{S}_1$ be the set of additional variables, and let $\mathbf{X}_{\mathcal{S}_{add}}$ be the corresponding random variable. The total set of observations can thus be expressed as the joint variable $\mathbf{X}_{\mathcal{S}_2} = (\mathbf{X}_{\mathcal{S}_1}, \mathbf{X}_{\mathcal{S}_{add}})$.*

**2.  Core Derivation via Conditional Mutual Information**   *Our objective is to prove that $H(\mathbf{w}|\mathbf{X}_{\mathcal{S}_1}, \mathbf{X}_{\mathcal{S}_{add}}) \leq H(\mathbf{w}|\mathbf{X}_{\mathcal{S}_1})$.*

*We begin by applying the chain rule for conditional entropy to $H(\mathbf{w}, \mathbf{X}_{\mathcal{S}_{add}}|\mathbf{X}_{\mathcal{S}_1})$ in two different ways:*

$$H(\mathbf{w}, \mathbf{X}_{\mathcal{S}_{add}}|\mathbf{X}_{\mathcal{S}_1}) = H(\mathbf{w}|\mathbf{X}_{\mathcal{S}_1}) + H(\mathbf{X}_{\mathcal{S}_{add}}|\mathbf{w}, \mathbf{X}_{\mathcal{S}_1}) \tag{11}$$

$$H(\mathbf{w}, \mathbf{X}_{\mathcal{S}_{add}}|\mathbf{X}_{\mathcal{S}_1}) = H(\mathbf{X}_{\mathcal{S}_{add}}|\mathbf{X}_{\mathcal{S}_1}) + H(\mathbf{w}|\mathbf{X}_{\mathcal{S}_1}, \mathbf{X}_{\mathcal{S}_{add}}) \tag{12}$$

*Equating the right-hand sides of equation 11 and equation 12, we get:*

$$H(\mathbf{w}|\mathbf{X}_{\mathcal{S}_1}) + H(\mathbf{X}_{\mathcal{S}_{add}}|\mathbf{w}, \mathbf{X}_{\mathcal{S}_1}) = H(\mathbf{X}_{\mathcal{S}_{add}}|\mathbf{X}_{\mathcal{S}_1}) + H(\mathbf{w}|\mathbf{X}_{\mathcal{S}_1}, \mathbf{X}_{\mathcal{S}_{add}}) \tag{13}$$

*Rearranging the terms to isolate the difference between the entropies of interest:*

$$H(\mathbf{w}|\mathbf{X}_{\mathcal{S}_1}) - H(\mathbf{w}|\mathbf{X}_{\mathcal{S}_1}, \mathbf{X}_{\mathcal{S}_{add}}) = H(\mathbf{X}_{\mathcal{S}_{add}}|\mathbf{X}_{\mathcal{S}_1}) - H(\mathbf{X}_{\mathcal{S}_{add}}|\mathbf{w}, \mathbf{X}_{\mathcal{S}_1}) \tag{14}$$

*The right-hand side of Equation equation 14 is, by definition, the conditional mutual information between $\mathbf{w}$ and $\mathbf{X}_{\mathcal{S}_{add}}$ given $\mathbf{X}_{\mathcal{S}_1}$:*

$$I(\mathbf{w}; \mathbf{X}_{\mathcal{S}_{add}}|\mathbf{X}_{\mathcal{S}_1}) = H(\mathbf{X}_{\mathcal{S}_{add}}|\mathbf{X}_{\mathcal{S}_1}) - H(\mathbf{X}_{\mathcal{S}_{add}}|\mathbf{w}, \mathbf{X}_{\mathcal{S}_1}) \tag{15}$$

**3. Non-Negativity of Conditional Mutual Information** *A fundamental theorem in information theory states that conditional mutual information is always non-negative. This can be rigorously shown by expressing it as an expected Kullback-Leibler (KL) divergence:*

$$I(\mathbf{w}; \mathbf{X}_{\mathcal{S}_{add}} | \mathbf{X}_{\mathcal{S}_1}) = \mathbb{E}_{p(\mathbf{x}_{\mathcal{S}_1})} \left[ D_{KL} \left( p(\mathbf{w}, \mathbf{x}_{\mathcal{S}_{add}} | \mathbf{x}_{\mathcal{S}_1}) \parallel p(\mathbf{w} | \mathbf{x}_{\mathcal{S}_1}) p(\mathbf{x}_{\mathcal{S}_{add}} | \mathbf{x}_{\mathcal{S}_1}) \right) \right] \tag{16}$$

*Since the KL divergence $D_{KL}(P \parallel Q) \geq 0$ for any two probability distributions $P$ and $Q$, the expectation of this non-negative quantity must also be non-negative. Thus,*

$$I(\mathbf{w}; \mathbf{X}_{\mathcal{S}_{add}} | \mathbf{X}_{\mathcal{S}_1}) \geq 0 \tag{17}$$

**4. Final Conclusion** *By substituting the mutual information definition from equation 15 back into equation 14 and applying the non-negativity property from equation 17, we obtain:*

$$H(\mathbf{w} | \mathbf{X}_{\mathcal{S}_1}) - H(\mathbf{w} | \mathbf{X}_{\mathcal{S}_1}, \mathbf{X}_{\mathcal{S}_{add}}) \geq 0 \tag{18}$$

*This directly implies the desired inequality:*

$$H(\mathbf{w} | \mathbf{X}_{\mathcal{S}_1}, \mathbf{X}_{\mathcal{S}_{add}}) \leq H(\mathbf{w} | \mathbf{X}_{\mathcal{S}_1}) \tag{19}$$

*Given that $\mathbf{X}_{\mathcal{S}_2} = (\mathbf{X}_{\mathcal{S}_1}, \mathbf{X}_{\mathcal{S}_{add}})$, the lemma is proven.*

**5. Condition for Equality** *Equality holds if and only if the conditional mutual information is zero, $I(\mathbf{w}; \mathbf{X}_{\mathcal{S}_{add}} | \mathbf{X}_{\mathcal{S}_1}) = 0$. As shown in equation 16, this occurs if and only if the joint conditional distribution factorizes into the product of the marginal conditional distributions:*

$$p(\mathbf{w}, \mathbf{x}_{\mathcal{S}_{add}} | \mathbf{x}_{\mathcal{S}_1}) = p(\mathbf{w} | \mathbf{x}_{\mathcal{S}_1}) p(\mathbf{x}_{\mathcal{S}_{add}} | \mathbf{x}_{\mathcal{S}_1}) \tag{20}$$

*This is the definition of conditional independence of $\mathbf{w}$ and $\mathbf{X}_{\mathcal{S}_{add}}$ given $\mathbf{X}_{\mathcal{S}_1}$. In a physically coupled system like the ocean, where all variables are intricately linked through underlying dynamical equations, this condition is generally not met. Therefore, the inequality is typically strict, meaning additional distinct observations strictly reduce the uncertainty.*

## C  ALGORITHM

---

**Algorithm 1** The `VISION` Framework for Reconstruction with Dynamic Prompting

---

**Input:**
**Require:** A subset of observations $X_S \in \mathbb{R}^{|S| \times H \times W}$
**Require:** A learnable codebook of prompt templates $\mathcal{C}_P = \{P_k\}_{k=1}^K$
**Require:** Model parameters $\theta$ (including UOA, SCP, and GSAO modules)

**Output:**
**Ensure:** Reconstructed vertical velocity field $\hat{w} \in \mathbb{R}^{3 \times H \times W}$

1: **function** VISION_RECONSTRUCT$(X_S, \mathcal{C}_P, \theta)$
$\qquad\qquad\qquad\qquad\qquad\qquad\quad\,\triangleright$ **— *Stage 1: Adaptive Observation Embedding* —**
2: $\qquad Z_0 \leftarrow \text{UOA}(X_S, m)$ $\qquad\quad\triangleright$ Canonicalize variable inputs into a fixed-dim feature map

$\qquad\qquad\qquad\qquad\qquad\qquad\quad\,\triangleright$ **— *Stage 2: State-Conditioned Prompt Generation* —**
3: $\qquad e \leftarrow \text{GlobalAveragePooling}(Z_0)$ $\qquad\qquad\quad\triangleright$ Compress $Z_0$ to a state vector
4: $\qquad v_{\text{context}} \leftarrow \text{Conv1}(e)$ $\qquad\qquad\triangleright$ Combine state and availability information
5: $\qquad \alpha \leftarrow \text{Softmax}(\text{MLP}_{\text{mixer}}(v_{\text{context}}))$ $\qquad\triangleright$ Generate mixing weights from context
6: $\qquad P \leftarrow \sum_{k=1}^K \alpha_k P_k$ $\qquad\qquad\qquad\triangleright$ Create dynamic prompt from codebook
7: $\qquad P \leftarrow \text{Upsample}(P)$ $\qquad\qquad\qquad\triangleright$ Match spatial dimensions with $Z_0$

$\qquad\qquad\qquad\qquad\qquad\qquad\quad\,\triangleright$ **— *Stage 3: Prompt-Guided Reconstruction* —**
8: $\qquad Z_1 \leftarrow \text{PromptInteraction}(Z_0, P)$ $\quad\triangleright$ Fuse base features with the dynamic prompt, e.g.,
$\qquad Z_0 + \text{Conv}(Z_0, P)$
9: $\qquad \hat{w} \leftarrow \text{GSAO}(Z_1)$ $\quad\triangleright$ Process conditioned features with the Geometry-Scale Aware Operator
$\qquad$ backbone

10: $\qquad$ **return** $\hat{w}$
11: **end function**

*Note: The entire model, parameterized by $\theta$, is trained end-to-end by minimizing a loss function (e.g., Smooth L1 Loss) between the prediction $\hat{w}$ and the ground-truth $w$. The training data consists of samples with varying availability observations.*

---

## D  BENCHMARK DETAILS

The $KD48$ benchmark used in this paper is derived from LLC4320, which is based on the global ocean simulation of MITgcm (Marshall et al., 1997). LLC4320 has a spatial resolution of 1/48° and a temporal resolution of 1h with 90 vertical levels. The LLC4320 simulation is initialized from the output of the Estimating the Circulation and Climate of the Ocean, Phase II (ECCO2), project (Menemenlis et al., 2008). The LLC4320 model is forced by the 6-hourly, 0.168 horizontal resolution ECMWF atmospheric reanalysis, as well as by an equivalent surface pressure field consisting of the full lunar and solar tidal potential (Weis et al. 2008). For our analysis, we select a regional near Kuroshio and use the hourly snapshots of sea surface height (SSH), sea surface potential temperature (SST), sea surface salinity (SSS), surface longitude velocity (U), surface latitude velocity (V), and depth level vertical velocity w at 20, 40, and 60, from 1 November 2011 to 31 October 2012 (366 days). For this regional data, height and width are both 512. Further, we use SSS and SST to calculate buoyancy (B) as a available observation, which can be expressed as:

$$b \;=\; g\Big[\alpha\big(\text{SST} - T_0\big) \;-\; \beta\big(\text{SSS} - S_0\big)\Big], \tag{21}$$

where, $b$ denotes the buoyancy, $g = 9.81 \text{ m s}^{-2}$ is the gravitational acceleration. The constants $\alpha$ (K$^{-1}$) and $\beta$ (psu$^{-1}$) are the thermal expansion and haline contraction coefficients, evaluated from the seawater equation of state at a chosen reference state $(T_0, S_0, p{=}0)$. Equation equation 21 is derived from the linearized equation of state, $\rho' \approx -\rho_0 \alpha(\text{SST} - T_0) + \rho_0 \beta(\text{SSS} - S_0)$, together

with $b = -g\rho'/\rho_0$. Given that limination that the observation obtained by available sensor may not reconstruct vertical velocity in the near- and superinertial bands. Rather than the full $w$ signals shown target, we will use the low-pass-filtered $w$ field:

$$\tilde{w}(t,z,x,y) = \sum_{\tau=t-L+1}^{t} \frac{1}{L} \exp\left(-\frac{t-\tau}{L}\right) w(\tau,z,x,y), \quad (22)$$

where, $w(t,z,x,y)$ denotes the vertical velocity at time $t$, depth $z$, and horizontal location $(x,y)$. The operator $\tilde{w}(t,z,x,y)$ is the temporally low-pass–filtered vertical velocity obtained through a causal exponential moving average with window length $L$. In our setup, L=1.2 day. The exponential weight $\exp(-(t-\tau)/L)$ emphasizes more recent states while progressively damping high-frequency variability.

## E EXPERIMENTS DETAILS

### E.1 TRAINING DETAILS

Since the vertical velocity $w$ has a much smaller magnitude compared to the input surface variables, it is essential to apply normalization to ensure stable learning. And different ocean variables have large variations in their magnitude. To allow the model focusing on reconstruction rather than learning the differences between variables, we normalized the data before feeding them into the model, so that the network can focus on reconstruction rather than being dominated by scale differences across variables. Specifically, we compute the mean and standard deviation of each variable from the training dataset, and use them to normalize the data. For a given variable, we subtract its corresponding mean and divide by its standard deviation, thereby mapping all variables to a comparable scale. We train all baselines and our `VISION` for 50 epochs with a learning rate 1e-4.

### E.2 EVALUATION METRIC

To comprehensively evaluate the reconstruction performance, we adopt three commonly used metrics: Root Mean Square Error (RMSE), Mean Absolute Error (MAE), and Pearson Correlation Coefficient (PCC).

**Root Mean Square Error (RMSE).** RMSE measures the square root of the average squared differences between predictions and ground truth, averaged across all samples, where larger errors contribute quadratically, making RMSE more sensitive to outliers:

$$\text{RMSE} = \frac{1}{B} \sum_{i=1}^{B} \sqrt{\frac{1}{M} \sum_{j=1}^{M} \left(p_{i,j} - t_{i,j}\right)^2}, \quad (23)$$

where, $B$ denotes the number of samples, $H$ and $W$ are the spatial dimensions (height and width). $M = H \times W$ is the total number of spatial locations per sample. $p_{i,j}$ is the predicted value at location $j$ of the $i$-th sample and $t_{i,j}$ is the ground truth value at location $j$ of the $i$-th sample.

**Mean Absolute Error (MAE).** MAE computes the mean of the absolute differences between reconstruction results and ground truth, which reflects the average error magnitude and is less sensitive to extreme values compared to RMSE:

$$\text{MAE} = \frac{1}{B} \sum_{i=1}^{B} \frac{1}{M} \sum_{j=1}^{M} |p_{i,j} - t_{i,j}|. \quad (24)$$

**Pearson Correlation Coefficient (PCC).** PCC evaluates the linear correlation between predicted and ground truth fields, which ranges from $-1$ (perfect negative correlation) to $+1$ (perfect positive correlation). A higher PCC indicates stronger alignment of spatial patterns between prediction and

ground truth:

$$\text{PCC} = \frac{1}{B} \sum_{i=1}^{B} \frac{\sum\limits_{j=1}^{M} p_{i,j} t_{i,j} - \frac{1}{M} \left(\sum\limits_{j=1}^{M} p_{i,j}\right) \left(\sum\limits_{j=1}^{M} t_{i,j}\right)}{\sqrt{\left(\sum\limits_{j=1}^{M} p_{i,j}^2 - \frac{1}{M}\left(\sum\limits_{j=1}^{M} p_{i,j}\right)^2\right)\left(\sum\limits_{j=1}^{M} t_{i,j}^2 - \frac{1}{M}\left(\sum\limits_{j=1}^{M} t_{i,j}\right)^2\right)}}. \tag{25}$$

