# OpenReview forum: "VISION: Prompting Ocean Vertical Velocity Reconstruction from Incomplete Observations"
_ICLR.cc/2026/Conference — ICLR 2026 Conference Withdrawn Submission_

### Official Review · Reviewer_eaeX · 2025-10-28

**Soundness:** 3
**Presentation:** 2
**Contribution:** 3
**Rating:** 4
**Confidence:** 5

**Summary:**

The authors propose VISION, a deep learning framework for reconstructing subsurface ocean vertical velocity (w) fields from incomplete surface observations. The paper tackles two issues in the domain, lack of standardized benchmarks and sensitivity to missing inputs.

**Strengths:**

The challenge of reconstructing subsurface fields from incomplete data is critical in physical oceanography. The proposed addressed a problem that is genuinely impactful, data uncertainty and robustness, particularly as the machine learning for earth sciences matures.

The authors contribute a benchmark KD48 which is large scale and can provide a step stone for further improvements in the field.

Qualitative and quantitative results showcase the effectiveness of the method.

**Weaknesses:**

Major weaknesses

1) There is a lot of nomeclature jargon that can be confusing while reading the paper. The use of prompting for example, in the paper may imply the use of VLMs, however the authors are using the term `prompting` refering to image inputs. Isn't the Geometry-Scale aware operator essentially a U-Net?

2) Lines 213-214: The way the dynamic projection is explained is confusing implying that the projection matrix is deformable and can handle arbitrary input dimensions. Does this section imply that the authors train multiple projection matrices for all cases of missing inputs? Since this creates a combinatorial problem does this mean that the same projection matrix is used for the same input number of dimensions?

3) The lemma from information theory is just a claim in the paper I believe it can be removed. There is no proof that additional data necessarily provide additional information and there is no experiment that clearly captures the feature importance.

4) "All experiments are conducted on 8 40GB Gpus": Is this cluster necessary also for inference?

5) Line 376: The baseline models are trained using complete observations but VISION is trained using random observation. Althouhg seamingly this means that VISION had to learn a harder task, the act of droping randomly observation can be consisdered augmentation which can contribute to the robustness of the network. For a fair comparison all methods should be compared under the same settings. To showcase the added robustness of VISION the authors can compare to the base VISION trained on CO.

**Questions:**

See weaknesses.

---

### Official Review · Reviewer_n2HH · 2025-10-31

**Soundness:** 4
**Presentation:** 3
**Contribution:** 1
**Rating:** 2
**Confidence:** 4

**Summary:**

This paper introduces an ocean vertical velocity reconstruction benchmark constructed from an existing dataset as well as a method to perform reconstruction from incomplete sea surface inputs by utilizing a module that standardizes any set of inputs into a prompt that their reconstruction module can accept as input. The paper shows improved quantitative performance compared to previous methods.

**Strengths:**

The benchmark seems reasonable and could encourage future work in this space, which is a valuable contribution.

**Weaknesses:**

It is not clear to me, despite the quantitative improvements over prior method, whether the proposed method produces better qualitative results useful for domain experts or those who might otherwise care about using the results in practice. In Figure 3, while the proposed method is clearly better than FNO and DDN, it is not clear to me visually whether their method is better than UNet, ResNet, or SimVP, and these methods are also not addressed by the figure's caption. I think it would be helpful if the authors more clearly described the actual structure of the subsurface ocean currents portrayed in the qualitative results, and explained how the higher quality structures reconstructed by their model can actually help the domain experts who might look at these reconstructions. For example, does higher quality reconstruction mean more accurate climate forecasting? And so on.

Additionally, it does not seem to me that the thoroughness of the ablations are commensurate with the complexity of the proposed model architecture. While the authors do ablate the entirety of the State Conditioned prompting (SCP) and the Geometry-Scale Aware Operator (GSAO) modules, these modules themselves are composed of complex submodules that don't get ablated. Additionally, I could not find any citations in the methods section that indicate the design of these modules is standard in the literature and already analyzed by other people. My feel is that for this kind of a task a much simpler architecture could suffice, even a vision transformer [1] where one associates the different kinds of inputs with particular positional encodings, and puts null tokens in the input where ever input variables are missing.

[1] https://arxiv.org/abs/2010.11929

**Questions:**

Main questions repeated from weaknesses section:

- How does the improved reconstruction quality translate into insight usable for real science informed decision making or modeling?
- What is the justification for the complex design of the SCP and GSAO submodules? More thorough ablations or references to literature suffice to answer this question.

---

### Official Review · Reviewer_pkDy · 2025-11-01

**Soundness:** 2
**Presentation:** 3
**Contribution:** 3
**Rating:** 4
**Confidence:** 4

**Summary:**

The authors construct KD48, a large-scale benchmark derived from simulation. The dataset includes five surface variables (SSH, SST, SSS, U, V) as inputs and three target depths (20, 40, 60 levels) of subsurface w. The paper also introduces VISION, a prompt-based framework for reconstructing subsurface ocean vertical velocity (w) from incomplete surface observations. The design allows the network to flexibly adapt to any subset of input variables without retraining. Experiments on KD48 demonstrate that VISION, even when trained with incomplete observations, outperforms state-of-the-art baselines trained with full inputs.

**Strengths:**

1. The paper makes a technical and infrastructural contribution to oceanographic machine learning by providing both a high-quality benchmark (KD48) and a well-designed model (VISION).
2. The Dynamic Prompting mechanism elegantly solves a key real-world challenge: missing and variable ocean observations.
3. The architecture design is conceptually clean and theoretically supported through information-theoretic reasoning about observational entropy reduction.

**Weaknesses:**

1. While the performance improvements are convincing, the reported metrics lack statistical uncertainty or significance testing across multiple runs.
2. The paper does not include a comparison with existing ocean machine learning datasets.
3. The training and computational cost of VISION (especially the dynamic prompting and deformable convolutions) are not discussed.
4. The study remains simulation-based and focuses on a single dataset. No validation is performed using real-world data.
5. The prompt design could be more thoroughly compared against simpler conditioning schemes to isolate its true added value.
6. The paper focuses only on reconstructing vertical velocity (w), even though w is strongly influenced by many other variables (beyond surface SSH, SST, SSS, U, V). Extending the framework to multiple variables would strengthen its general impact.

**Questions:**

1. How sensitive is the model’s performance to the choice of prompt codebook size and embedding dimension?
2. Could the same prompting principle be applied to temporal prediction, not just static reconstruction?
3. How do you ensure that the filtered w retains physically meaningful variance and avoids suppressing real submesoscale signals?

---

### Note · Authors · 2026-01-12

I have read and agree with the venue's withdrawal policy on behalf of myself and my co-authors.